# Development and Psychometric Assessment of a Questionnaire for the Detection of Invisible Violence against Women

**DOI:** 10.3390/ijerph191711127

**Published:** 2022-09-05

**Authors:** Iria Dobarrio-Sanz, Anabel Fernández-Vargas, Alba Fernández-Férez, Diana Patricia Vanegas-Coveña, Otilia Vanessa Cordero-Ahiman, José Granero-Molina, Cayetano Fernández-Sola, José Manuel Hernández-Padilla

**Affiliations:** 1Department of Nursing, Physiotherapy and Medicine, University of Almeria, 04120 Almeria, Spain; 2Unidad de Cuidados Intensivos, Hospital Universitario Torrecárdenas, 04009 Almeria, Spain; 3Distrito Sanitario Almería, 04009 Almeria, Spain; 4Facultad de Ciencias Médicas, Universidad de Cuenca, Cuenca 010107, Ecuador; 5Grupo de Investigación en Economía Regional (GIER), Facultad de Ciencias Económicas y Administrativas, Universidad de Cuenca, Cuenca 010107, Ecuador; 6Facultad de Ciencias de la Salud, Universidad Autónoma de Chile, Santiago 7500000, Chile

**Keywords:** health service environment, intimate partner violence, invisible sexism, prevention questionnaire, validation study

## Abstract

Background. Invisible violence against women (IVAW) can be understood as the set of attitudes, behaviors, and subtle beliefs that men use to subordinate women and that are culturally accepted. These behaviors can be a risk factor for intimate partner violence (IPV), so it is important to design tools that allow us to detect it early. The aim of this study was to design and psychometrically assess a questionnaire for the detection of invisible violence against women (Q-IVAW). Methodology. A descriptive cross-sectional methodological study carried out in three phases: (1) development of the initial version; (2) pilot study (*N* = 51); and (3) final validation study (*N* = 990). The tool’s reliability, validity, and legibility were assessed. To assess reliability, the internal consistency (Cronbach’s α) was analyzed. The validity assessment included an analysis of content, criterion, and construct validity. Results. The EFA revealed that the Q-IVAW was comprised of five factors that explained 55.85% of the total variance found. The Q-IVAW showed very high reliability (α = 0.937), excellent content validity, and good construct validity. The criterion validity analysis showed a moderate correlation between A-IPVAW and Q-IVAW (*r* = 0.30; *p* < 0.001). Conclusion. The psychometric assessment of the Q-IVAW yielded good results, which could support the tool’s ability to assess how often women are subjected to inviable violent behaviors by their partners.

## 1. Introduction

Intimate partner violence (IPV) is the most common form of violence experienced by women [1,2]. IPV consists of physical, psychological, and/or sexual abuse of a woman by her current or former partner [3]. It is estimated that almost 30% of women worldwide have suffered some form of IPV [4]. In Spain, around 24% of women of all ages have suffered IPV [4], and this figure rises to 43% in specific age groups [5]. IPV is associated with worse biopsychosocial health for both women [6,7,8] and their relatives [9,10,11,12,13,14]. IPV poses a socioeconomic challenge to many countries and their healthcare systems [15,16,17]. Although women who suffer IPV attend primary care clinics and emergency care units more frequently [7,18,19,20,21], they do not report having been abused by their partners [22,23] and do not present visible signs that would facilitate IPV detection [24,25,26,27]. Consequently, healthcare professionals face the challenge of detecting cases of invisible violence against women as a form of IPV [16,26,28].

### Background

IPV has negative consequences on women’s physical health, and it is associated with musculoskeletal injuries [29]; chronic pain and fatigue [30,31,32]; malnutrition [33]; gastrointestinal, gynecological, and sexual [6,7,34,35] disorders; increased cardiovascular risk [36,37]; and increased mortality [38]. Recently, IPV has also been associated with an increased risk of COVID-19 infection [39]. IPV negatively impacts mental health, and it is associated with increased levels of anxiety and depression [7,8,40], post-traumatic stress disorder, substance abuse [7,41], and suicidal ideation [7,42,43,44]. In addition, IPV is associated with poorer quality of life [8], social isolation [42,45], deterioration of relationships with family and friends [46], job instability [47], and deterioration of personal values and beliefs [48,49,50].

Invisible violence against women (IVAW) is a part of IPV and it can be understood as culturally accepted attitudes, behaviors, and subtle beliefs that men use to subordinate women [51,52,53,54]. IVAW is the legitimization of gender stereotypes, discrimination against women for being women, and the use of coercive and hostile behaviors to achieve power and domination [55,56,57]. From a theoretical point of view, it is understood that everyday social life is profoundly influenced by gender inequalities [58]. IVAW is a form of symbolic violence that has been naturalized and incorporated into the social habitus [59,60], becoming difficult to recognize even by women who are exposed to this type of violence [58]. IVAW is based on behaviors of ambivalent sexism and sexist microaggressions that reflect the daily domination exercised by men over women [51,52,53,54,61].

Several authors defend the idea that IVAW and its acceptability can lead to more explicit and severe forms of IPV [62,63,64,65]. In fact, the WHO urges us to highlight gender attitudes and stereotypes that subjugate women and perpetuate male privilege, justifying violence against them [22]. Therefore, the early detection of IVAW becomes a cornerstone in the prevention of more explicit forms of IPV and healthcare professionals are key in this process [66]. However, healthcare professionals have difficulties in providing adequate care to women who suffer from IVAW as a form of IPV [28,67,68]. Consequently, the development of standardized tools could help healthcare professionals to prevent explicit forms of IPV by identifying cases of IVAW early on [24,69,70]. The aim of the study was to design and psychometrically assess a questionnaire for the detection of invisible violence against women (Q-IVAW).

## 2. Materials and Methods

### 2.1. Design

A descriptive cross-sectional methodological study for the development of the Q-IVAW.

### 2.2. Study Participants and Sampling

The study was designed and conducted at the University of Almeria in Spain (UAL). Participants were recruited using a convenience sampling method. Participants who met the following criteria were included: (1) to identify oneself as a woman, (2) to be at least 18 years old, (3) to have a male partner, and (4) to agree to participate voluntarily in the study. The only exclusion criterion was not being able to read. A sample of 51 women was recruited for the pilot study [71], and they did not participate in the final validation study. On the other hand, the sample for the final validation study had to meet the criterion of being comprised of at least 10 subjects for each scale item [71,72].

### 2.3. Data Collection

The data were collected using an online questionnaire that was distributed through several social media platforms and women’s associations. Despite all the items of the questionnaire being compulsory to complete, we included the response option “I would prefer not to disclose this information” for all the sociodemographic questions. This tried to make women feel more comfortable to participate honestly without fearing being identifiable. Furthermore, the email addresses of the main researchers were provided in the questionnaire for women to contact us if they needed clarification with any of the items. This form was distributed in Spanish, so the results refer to the Spanish version of the questionnaire. The data collection process was carried out from September 2021 to January 2022.

### 2.4. Ethical Aspects

The Ethics and Research Committee of the Department of Nursing, Physiotherapy and Medicine of the UAL granted approval of the research project (EFM 141/2021). The participants received information about the study (justification, objective, methods, their rights as participants, right to confidentiality and anonymity), and all gave their written consent to participate before completing the questionnaire.

### 2.5. PHASE 1: Item Development and Pilot Study of the Q-IVAW

#### 2.5.1. Item Development

The items were developed by the research team and reflected behaviors associated with IVAW according to Bonino’s Theory of Sexist Microaggressions [51,52,53] and Glick and Fiske’s Theory of Ambivalent Sexism [54]. Subsequently, the questionnaire was evaluated by a panel of 24 experts (5 nurses, 5 physicians, 8 psychologists, 3 social workers, and 3 teachers) with more than 10 years of experience working with survivors of intimate partner violence. The initial pilot version of the Q-IVAM included 33 items with 5 response options (0 = never; 1 = rarely; 2 = sometimes; 3 = often; 4 = always) that measured the frequency with which the woman’s partner exhibited behaviors labelled as invisibly violent against them.

#### 2.5.2. Pilot Study Methods

For the pilot study, the Q-IVAW was tested for content validity and reliability.

##### Validity

Content validity in the pilot version of the Q-IVAW was explored by calculating the content validity index of each item (I-CVI). Content validity was calculated from the individual rating that the experts gave to each individual item: 1 = irrelevant; 2 = not very relevant; 3 = quite relevant; 4 = very relevant [73] to assess when a woman was being exposed to IVAW. An I-CVI ≥ 0.78 was considered acceptable when it was evaluated by 15 or more experts [73].

##### Reliability

To analyze the internal consistency of the Q-IVAW, Cronbach’s α of the scale, the corrected item-total correlation index (CCIT), and Cronbach’s α of the scale when one item was removed were calculated [72,73,74]. Items were removed from the Q-IVAW if they met both of the following criteria: [1] to have an IT-CCI lower than 0.3, and [2] the questionnaire’s Cronbach’s α increased significantly when that item was removed. A Cronbach’s α greater than 0.7 was considered indicative of good internal consistency.

#### 2.5.3. Results of the Pilot Study

The results of the pilot study are shown in Table 1. The CVI of all the items was higher than 0.78 so all were retained and administered to the pilot sample (*N* = 51). The Cronbach’s α of the Q-IVAW was 0.944. Item 4 and item 6 did not meet the criteria to be retained as part of the Q-IVAW. Consequently, 31 items of the Q-IVAW were retained for the final validation study.

### 2.6. PHASE 2: Final Psychometric Validation Study of the Q-IVAW

The psychometric analysis for the final validation study was carried out following the methods and recommendations of expert authors in this methodology [72,73,75]. The statistical analysis was performed with IBM^®^ SPSS Statistics^®^ 27 software (IBM, Armonk, NY, USA).

#### 2.6.1. Validity of the Q-IVAW in the Final Validation Study

For the final validation of the Q-IVAW, its content validity, criterion validity, and construct validity were explored.

##### Content Validity

The content validity of the final validation study of the Q-IVAW was assessed using the same method as that described in the pilot study. Likewise, the total average CVI (t-CVI) of the scale was calculated and was considered appropriate if it was equal to or greater than 0.78 [73].

##### Criterion Validity

Criterion validity was examined by correlating participants’ scores on the Q-IVAW and the Acceptability of Intimate Partner Violence Against Women (A-IPVAW) scale [76]. Although the A-IPVAW is based on attitudes of acceptability of more explicit and severe IPV behaviors than those included on the Q-IVAW, the relationship between ambivalent behaviors and acceptability of more severe IPV has been previously evidenced [76]. Prior to this, we performed a normality test to the mean Q-IVAW score: kurtosis (value = 3.60; SE =0.66), skewness (value = 1.61; SE = 0.33), and Shapiro–Wilk test (W (51) = 0.87; *p* < 0.001). Since it did not follow a normal distribution, Spearman’s correlation coefficient (r) was calculated between the scores on the Q-IVAW and the A-IPVAW.

##### Construct Validity

Construct validity was studied by means of an exploratory factor analysis (EFA) with a principal axis factorization matrix and Varimax rotation. Adequacy tests for this analysis were conducted prior to conducting the EFA. The EFA was considered an appropriate analysis if Bartlett’s test was significant (*p* < 0.05) and the Kaiser Meyer Olkim index (KMO) was greater than 0.7 [72,73,77]. Similarly, those factors that met three criteria were considered dimensions of the scale: [1] having an eigenvalue greater than 1, [2] including items with a factor loading ≥ 0.40, and [3] having a clear graphical representation in the eigenvalue plot [72,77]. Items that did not clearly load on any of the factors (factor loading < 0.40) were deleted from the Q-IVAW. To complete the construct validity analysis of the Q-IVAW, a known-groups analysis (KGA) was performed and the scores of the participants were compared using the Kruskal–Wallis for K independent samples.

#### 2.6.2. Reliability of the Q-IVAW in the Final Validation Study

The method followed to assess the reliability of the Q-IVAW was the same as the one described in the pilot study.

### 2.7. Readability

Readability was assessed using the INFLESZ scale, which categorizes a text in 5 levels of difficulty ranging from 0 to 100 points: (<40) = very difficult; (40–55) = somewhat difficult; (55–65) = normal; (65–80) = fairly easy; (>80) = very easy [78]. In addition, the time the participants dedicated to read and complete the scale was collected.

### 2.8. Q-IVAW Scoring System

An internal scoring system with 4 categories was developed [79]: [1] scores of ≥1 standard deviation (*SD*) below the mean; [2] scores of ≤2 *SD* above the mean; [3] scores of ≤4 *SD* above the mean; and [4] scores of >4 *SD* above the mean.

## 3. Results

### 3.1. Description of the Sample

The sociodemographic characteristics of the sample participating in the final validation study (*N* = 990) are shown in Table 2. The mean age in years was 34.36 (*SD* = 11.66) and 60.9% of participants (*n* = 603) considered themselves a feminist. In addition, 54.5% (*n* = 540) had completed either primary or secondary education and 35.5% (*n* = 351) declared having an average individual income (1000–3000 €). Other relevant sociodemographic information can be seen in Table 2.

### 3.2. Validity

#### 3.2.1. Content Validity

In terms of content validity, the I-CVI values were above 0.78 for all items (range 0.88–1). The t-CVI of the Q-IVAW was 0.96 and was therefore considered excellent (Table 3).

#### 3.2.2. Criterion Validity

The analysis of criterion validity showed a moderate degree of correlation (*r* = 0.3; *p* < 0.001) between the mean score of the Q-IVAW and the A-IPVAW.

#### 3.2.3. Construct Validity

The results of the construct validity analysis are presented below.

##### Exploratory Factor Analysis

The Barlett’s test of sphericity (*X*^2^(276) = 4263.62; *p* < 0.001) and the Kaiser Meyer Olkin test (KMO = 0.93) indicated that it was pertinent to perform the exploratory factor analysis (EFA). After performing four rounds of exploratory factor analyses (EFA), items 3, 7, 8, 11, 12, 26, 27, and 29 were deleted (factor loading < 0.40). An EFA of the resulting 23-item version of the Q-IVAW extracted 5 factors that accounted for 55.83% of the total variance explained. The 23 items comprising the Q-IVAW only loaded onto one factor (factor loading ≥ 0.40). The EFA results are shown in Table 3. The final version of the questionnaire can be requested from the authors.

##### Known Groups Analysis

The results of the KGA showed statistically significant differences between women’s scores depending on their partner’s income (*X*^2^(2) = 11.48; *p* = 0.005). However, when the participants’ scores for each subscale were tested individually, we found that only the subscales “utilitarian sexist behaviours” and “coercive sexist behaviours” subscales, together with the total Q-IVAW’s score, yielded significant differences amongst groups (see Table 4). No significant differences were found with respect to other sociodemographic characteristics.

### 3.3. Reliability of the Scale

The reliability analysis of the final version of the Q-IVAW is shown in Table 5. All the items showed an CCIT > 0.3 and the Cronbach’s α of the scale would not have increased if any item had been deleted. The final 23-item version of the Q-IVAW showed excellent internal consistency (α = 0.937).

### 3.4. Readability

According to the INFLESZ scale [78], the readability level of the Q-IVAW is “fairly easy” (73.04 points). The participants did not report any comprehension problems, so no modifications were made to the scale in this regard. The average time for reading and completing the scale was approximately 6.5 min.

### 3.5. Q-IVAW Scoring System

The mean total score of the Q-IVAW was 17 (*SD* = 16). Four categories of interpretation were created according to how often participants were subjected to invisible violence against women by their partners [79]: hardly ever = 0–17 points (≥1 *SD* below the mean); occasionally = 18–49 points (≤2 *SD* above the mean); often = 50–80 points (≤4 *SD* above the mean); and very often = 81–92 points (>4 *SD* above the mean).

## 4. Discussion

The purpose of this study was to design and psychometrically assess a questionnaire for the detection of invisible violence against women (Q-IVAW). Bonino’s theory of sexist microaggressions [51,53] and Glick and Flick’s theory of ambivalent sexism [54] were used as theoretical models to develop the tool. The questionnaire’s psychometric properties were tested following the recommendations by Coaley [72], Polit and Beck [73], and Streiner and Kottner [75]. Consequently, the reliability and validity of the questionnaire were assessed. The readability of the questionnaire was also examined to assess whether its content was readable and understandable by the target population.

In the initial stage of the study, the questionnaire was comprised of 33 items. After conducting a pilot study, two items (item 4 and item 6) were deleted because they did not meet the reliability criteria to keep them [72,73,75]. Once these two items were removed from the Q-IVAW, the pilot version of the questionnaire showed high internal consistency and was administered to a larger population in the final validation study (*N* = 990).

After performing four rounds of EFA in the final validation stage, eight items (item 3, item 7, item 8, item 11, item 12, item 26, item 27, and item 29) were removed from the Q-IVAW because they did not load on any of the factors extracted. The final 23-item version of the Q-IVAW showed excellent internal consistency, which is understood as evidence of good reliability [71,72]. The Q-IVAW’s content validity after expert evaluation yielded excellent results and suggests that all the items contribute to measuring the frequency with which participants’ partners engage in invisible violence against women [51,52,53,54]. Regarding criterion validity and due to the lack of validated tools measuring the same construct as the Q-IVAW, the Spanish version of the A-IPVAW was used for comparisons [63,76]. The A-IPVAW is based on explicit violence against women and the Q-IVAW includes more subtle violent behaviors; this may explain why the correlation between the two instruments was moderate. Construct validity was assessed using an exploratory factor analysis (EFA) and known-groups analysis (KGA). Five subdimensions emerged from the EFA: (1) crisis sexist behaviors; (2) utilitarian sexist behaviors; (3) coercive sexist behaviors; (4) ambivalent sexist behaviors; and (5) benevolent sexist behaviors. These subdimensions confirm the two theoretical models upon which the Q-IVAW was developed [51,52,53,54]. In relation to the known-group analysis conducted, the Q-IVAW could detect statistically significant differences in women’s scores depending on their partners’ income. According to these results, participants whose partners declared to have a high income were subjected to invisible violent behaviors more often. Although these results differ from those of previous studies [80,81], our findings are in line with the resource theory of power and its relationship with the theory of patriarchal norms [56]: if men earn more money, they will have more power within the relationship. Nevertheless, it is important to highlight that although the socioeconomic level of women and their partners is a key factor to take into account when designing interventions for the prevention and detection of IPV, some authors urge us to also focus on other cultural aspects, gender attitudes, and inequalities [82,83], which define the norms and values that govern interpersonal relationships [61,84,85].

### Limitations

This study has some limitations that need to be considered. The first limitation is that it was carried out in a specific sociocultural context, so further studies in other cultural settings would be necessary to be able to generalize the findings. Another limitation of the present study is related to the convenience sampling method used, which also limits the generalizability of the results. Online recruitment may make it difficult to verify the data provided in comparison with traditional methods (i.e., we cannot ensure that all the respondents were women meeting the inclusion criteria) [86]. In addition, the way the sample was recruited could have led to a self-choice bias, which means that those women who chose to participate in the study could be more aware, and therefore less tolerant, of this type of violence against women. In addition, due to organizational constraints, the temporal stability of the Q-IVAW was not examined, nor was a confirmatory analysis performed. It is important that future studies test temporal stability and conduct a confirmatory factor analysis on larger datasets of participants recruited through probabilistic sampling methods. Last, although the Q-IVAW can assess how often women are subjected to invisible violent behaviors from their partners, it is unclear whether its items and subdimensions would allow us to understand how women experience the phenomenon of invisible sexist violence as an indivisible whole. Validity and reliability are continuous, infinite, and incremental processes, so that, depending on the context in which the questionnaire is used, its psychometric properties will have to be established first.

## 5. Conclusions

Following a rigorous psychometric evaluation, the Q-IVAW showed good psychometric properties. Our results indicate that the Q-IVAW is a reliable and valid tool to measure invisible violence against women when this is perpetrated by their partners. The use of the Q-IVAW could help healthcare professionals to flag and follow-up cases in which women are often exposed to invisible violent behaviors from their partners as part of the prevention of more explicit forms of IPV. Future studies should focus on testing the Q-IVAW’s psychometric properties in different populations and in different sociocultural contexts to confirm the dimensionality of this tool.

## Figures and Tables

**Table 1 ijerph-19-11127-t001:** Psychometric properties of the pilot version of the Q-IVAW (*N* = 51).

	I-CVI *	Cronbach Alpha If the Item Is Deleted	CCIT **
*Please indicate how often your (male) partner engages in the following behaviors:*			
1. My partner makes sexual jokes (e.g., about the number of people he has had sex with, about rape, or about sexual preferences).	0.92	0.944	0.255
2. My partner makes jokes about gender stereotypes (e.g., about women’s abilities to do certain jobs, about women’s ability to drive, or their nature to perform household duties).	1	0.943	0.438
3. My partner tells me he wants to know where I am at all times, for safety reasons.	1	0.943	0.386
4. My partner tells me he wants to have access to my cell phone and its content.	1	0.947	0.147 **^†^**
5. My partner makes comments about the bodies of women who appear in adverts.	0.79	0.943	0.414
6. My partner openly says it is normal that women star in ads for detergent, household appliances, etc.	0.92	0.948	−0.003 **^†^**
7. My partner says that women are not as qualified as men to do some jobs (for example: miner, firefighter, army).	0.96	0.941	0.690
8. My partner is against favoring women to achieve equity in managerial positions.	0.96	0.946	0.217
9. My partner tries to impose his opinion to make decisions on issues that men know best (e.g., to buy a car).	0.96	0.940	0.751
10. There are things that my partner prefers me to do because I am a woman (for example: decorating the house, taking care of loved ones, cooking, or dealing with male salespeople to get a discount).	1	0.941	0.635
11. My partner only uses the masculine term when talking about men and women in plural (e.g., he says “actors” when referring to “actors” and “actresses”).	0.88	0.944	0.465
12. My partner uses sexist language to talk to me, even if he does not realize it.	0.96	0.941	0.721
13. My partner uses his physique (i.e., gestures, postures, etc.) or voice to impose his opinions when we argue.	1	0.941	0.742
14. My partner tends to want to take charge because he is a man.	0.96	0.940	0.795
15. My partner uses his manly logic as if it were the only right way to do things (e.g., he believes that childcare leave should be taken by the woman).	0.96	0.940	0.780
16. My partner insists until he gets what he wants even though I repeatedly make it clear that I do not agree.	0.96	0.941	0.689
17. My partner tells me that my housework has no economic value.	1	0.943	0.475
18. My partner tells me that I am more capable of caring for others just because I am a woman.	0.96	0.939	0.802
19. My partner tells me it is logical that I should be the one to look after the children or other loved ones (now or in the future).	0.96	0.940	0.768
20. My partner uses emotional blackmailing to get me to do something he wants.	0.92	0.941	0.681
21. My partner tends to overrule me when he disagrees with me.	1	0.941	0.757
22. My partner tells me that he makes decisions without consulting me as a way of protecting me or my family.	0.92	0.944	0.309
23. My partner crosses a line to protect me without consulting me first.	1	0.943	0.479
24. My partner plays hard to get in order to get things from me.	0.92	0.941	0.678
25. My partner withholds information from me to avoid what he believes to be unnecessary conflicts.	0.88	0.941	0.653
26. My partner flatters me to get what he wants.	0.88	0.943	0.510
27. My partner compares himself to other men to make me see that he is doing things right.	1	0.941	0.707
28. My partner pretends to be clueless (e.g., saying “I didn’t notice”) to justify certain harmful behaviors towards me.	1	0.940	0.774
29. My partner gives me moral support, but he does not take on more responsibilities when I need it.	1	0.941	0.678
30. My partner lets me make mistakes even though he knows I am not doing something correctly so he can reproach me for it afterwards.	1	0.941	0.805
31. My partner gives me gifts or promises in order to obtain some benefit.	0.96	0.941	0.703
32. My partner only gives in during disputes in order to get more benefits later.	1	0.942	0.669
33. My partner tries to make me feel sorry for him when feeling ill so that I look after him.	1	0.942	0.580

* Content validity index of each item. ** Item-total corrected correlation index. † Deleted item.

**Table 2 ijerph-19-11127-t002:** Sociodemographic information from the participants in the final validation study (*N* = 990).

Characteristics	Sample (*N* = 990)
	* n* (%)
Age (in years)	34.36 ± 11.66 *
Feminist	
Yes	603 (60.9)
No	387 (39.1)
Religion	
Christian	642 (64.8)
Not religious	333 (33.6)
Muslim	9 (0.9)
Others	6 (0.6)
Relationship duration	
Fewer than 3 months	14 (4.2)
3 months–1 year	30 (9.1)
1–3 years	65 (19.7)
3–10 years	101 (30.6)
10–20 years >20 years	64 (19.4) 56 (17.0)
Level of education completed	
University education	450 (45.4)
Secondary education	396 (40.0)
Primary education	144 (14.6)
Monthly women’s individual monthly income	
Low (less than €1000)	48 (4.8)
Average (€1000–3000)	351 (35.5)
High (more than €3000) Preferred not to say	294 (29.7) 297 (29.4)
Partner’s religion	
Christian	534 (53.9)
Not religious	438 (44.2)
Muslim	6 (0.6)
Others	12 (1.2)
Partner’s level of education	
University education	351 (35.5)
Secondary education	303 (30.6)
Primary education	336 (33.9)
Partner’s monthly income	
Low (less than €1000)	102 (10.1)
Average (€1000–3000)	618 (61.2)
High (more than €3000)	270 (26.7)

* Mean ± standard deviation (*SD*).

**Table 3 ijerph-19-11127-t003:** Results of the exploratory factor analysis of the Q-IVAW (*N* = 990).

ÍTEMS	FACTOR				
	1	2	3	4	5
**Crisis sexist behaviors**					
IT-1. My partner insists until he gets what he wants even though I repeatedly make clear that I do not agree.	0.529	0.153	0.211	0.181	0.247
IT-2. My partner uses emotional blackmailing to get me to do something he wants.	0.614	0.206	0.212	0.356	0.154
IT-3. My partner plays hard to get in order to get things from me.	0.567	0.158	0.218	0.267	0.394
IT-4. My partner lets me make mistakes even though he knows I am not doing something correctly so he can reproach me for it afterwards.	0.449	0.319	0.243	0.363	0.142
IT-5. My partner gives me gifts or promises in order to obtain some benefit.	0.738	0.170	0.152	0.082	0.127
IT-6. My partner only gives in during disputes in order to get more benefits later.	0.803	0.207	0.163	0.187	0.086
IT-7. My partner tries to make me feel sorry for him when feeling ill so that I look after him.	0.619	0.203	0.194	0.100	0.137
**Utilitarian sexist behaviors**					
IT-8. There are things that my partner prefers me to do because I am a woman (e.g., decorating the house, taking care of loved ones, cooking, or dealing with male salespeople to get a discount).	0.080	0.598	0.267	0.115	0.182
IT-9. My partner uses his manly logic as if it were the only right way to do things (e.g., he believes that childcare leave should be taken by the woman).	0.325	0.596	0.320	0.268	0.125
IT-10. My partner tells me that my housework has no economic value.	0.156	0.525	0.168	0.217	−0.092
IT-11. My partner tells me that I am more capable of caring for others just because I am a woman.	0.265	0.784	0.135	0.121	0.202
IT-12. My partner tells me it is logical that I should be the one to look after the children or other loved ones (now or in the future).	0.245	0.788	0.130	0.211	0.176
**Coercive sexist behaviors**					
IT-13. My partner tries to impose his opinion to make decisions on issues that men know best (e.g., to buy a car).	0.282	0.265	0.588	0.234	0.087
IT-14. My partner uses his physique (i.e., gestures, postures, etc.) or voice to impose his opinions when we argue.	0.315	0.248	0.535	0.195	0.188
IT-15. My partner tends to want to take charge because he is a man.	0.292	0.271	0.729	0.219	0.165
IT-16. My partner tends to overrule me when he disagrees with me.	0.345	0.241	0.567	0.301	0.188
**Ambivalent sexist behaviors**					
IT-17. My partner makes sexual jokes (e.g., about the number of people he has had sex with, about rape, or about sexual preferences).	0.209	0.080	0.130	0.580	0.111
IT-18. My partner makes jokes about gender stereotypes (e.g., about women’s abilities to do certain jobs, about women’s ability to drive, or their nature to perform household duties).	0.162	0.240	0.138	0.466	0.009
IT-19. My partner makes comments about the bodies of women who appear in adverts.	0.245	0.135	0.133	0.586	0.062
IT-20. My partner withholds information from me to avoid what he believes to be unnecessary conflicts.	0.309	0.231	0.171	0.420	0.307
IT-21. My partner pretends to be clueless (e.g., saying “I didn’t notice”) to justify certain harmful behaviors towards me.	0.264	0.244	0.306	0.487	0.070
**Benevolent sexist behaviors**					
IT-22. My partner tells me that he makes decisions without consulting me as a way of protecting me or my family.	0.398	0.216	0.212	0.258	0.457
IT-23. My partner crosses a line to protect me without consulting me first.	0.228	0.131	0.140	0.059	0.661
% of variance	17.516	13.260	9.763	9.638	5.648
% of accumulated variance	17.516	30.776	40.539	50.177	55.825

**Table 4 ijerph-19-11127-t004:** Known-group analysis results for the variable “partner’s monthly income”.

	Low Income (*n* = 102)	Average Income (*n* = 618)	High Income (*n* = 270)	Known-Group Differences *
	Mean Rank	Mean Rank	Mean Rank	*p*-Value
Utilitarian sexist behaviors	178.87	152.14	211.04	0.001
Coercive sexist behaviors	166.31	157.95	209.07	0.013
Total Q-IVAW	172.29	155.07	210.69	0.005

* The Kruskal–Wallis test was used.

**Table 5 ijerph-19-11127-t005:** Content validity index and reliability analysis of the 23-item Q-IVAW (*N* = 990).

	I-CVI *	Cronbach Alpha If the Item Is Deleted	CCIT **
*Please indicate how often your (male) partner engages in the following behaviors:*			
IT-1	0.96	0.934	0.588
IT-2	0.92	0.932	0.707
IT-3	0.92	0.933	0.690
IT-4	1	0.933	0.685
IT-5	0.96	0.934	0.609
IT-6	1	0.933	0.691
IT-7	1	0.934	0.590
IT-8	1	0.935	0.535
IT-9	0.96	0.932	0.735
IT-10	1	0.936	0.463
IT-11	0.96	0.933	0.667
IT-12	0.96	0.933	0.686
IT-13	0.96	0.934	0.581
IT-14	1	0.933	0.636
IT-15	0.96	0.932	0.709
IT-16	1	0.931	0.763
IT-17	0.92	0.937	0.438
IT-18	1	0.936	0.464
IT-19	0.79	0.936	0.454
IT-20	0.88	0.937	0.388
IT-21	1	0.932	0.712
IT-22	0.92	0.933	0.639
IT-23	1	0.937	0.436

* Content validity index of each item. ** Item-total corrected correlation index.

## Data Availability

Data are available from the authors (I.D.-S., A.F.-V. and J.M.H.-P.).

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
