# Peer review of "Development and Psychometric Assessment of a Questionnaire for the Detection of Invisible Violence against Women"

_ijerph, 2022, doi:10.3390/ijerph191711127_

Round 1

Reviewer 1 Report

The article is clearly formulated so that no suggestions for the authors are needed and the article is ready for print in its present form.

I highly appreciate that the subject is dealt with in a non-Mulim society because prejudices say that violence against women on behalf of their current or past partners be a pecularity of Islamic societies. It would, therefore, be interesting having parallel studies from other European societies because it seems that Spain is a specific case and I wonder why if so or that Spain is no exception to the rule regarding other European societies.

Author Response

Reviewer 1.

Reviewer’s comment: The article is clearly formulated so that no suggestions for the authors are needed and the article is ready for print in its present form.

I highly appreciate that the subject is dealt with in a non-Mulim society because prejudices say that violence against women on behalf of their current or past partners be a pecularity of Islamic societies. It would, therefore, be interesting having parallel studies from other European societies because it seems that Spain is a specific case and I wonder why if so or that Spain is no exception to the rule regarding other European societies.

Response: Thank you for your insight. We agree with you in that violence against women, and particularly invisible intimate partner violence, is a worldwide issue that needs to be addressed and evaluated from many different perspectives and cultural environments.

Recent governmental efforts in Spain have led to a series of laws and regulations that intend to protect women against not only explicit forms of IPV but also more subtle forms of such violence. This has opened many public debates and it is contributing to raise the awareness about the importance of not accepting culturally imbedded behaviours that can be considered sexist microaggressions. Not all European countries are at this stage of their fight against intimate partner violence and, therefore, we fully agree that more research is needed in the European context. If translated to other languages, the Q-IVAW could be tested and adapted to other European contexts so it could also be used to assess how many women are subjected to invisible violent behaviours from their partners.

Following your comment, we have included a comment in our paper to explicitly express the need for further research in other European countries.

Reviewer 2 Report

The introduction section is well justified and shows the relationship of the work with the relevant literature on IPV and its consequences. It correctly defines the objective of the study and the interest that a IVAW rapid detection questionnaire brings to healthcare professionals.

The materials and methods section includes the selection criteria and the sample sizes used. It is an online survey sent to social media platforms and women's associations, but,

1.- It does not indicate whether any response control strategy has been used, a characteristic that may influence the results obtained.

2.- It would also be useful to indicate if the surveys have been answered with the questionnaire written in Spanish and the items have been subsequently translated; the psychometric results obtained will refer to the version used, and may be different in other languages.

In the pretest reliability study (2.5.2.2) it is indicated that to retain an item it must have an IT-CCI equal to or greater than 0.3; however,

3.- In Table 1, items 1 and 8 would not meet this criterion. It would be necessary to modify the criterion in section 2.5.2.2 (for example, IT-ICC equal to or greater than 0.2) or eliminate both items.

4.- In the sociodemographic characteristics included in Table 2, it should be indicated whether the monthly income belongs to the person surveyed or to the household as a whole (the marital status of the people is not indicated either) since the economic dependence or independence can influence the results of perception of violence.

5.- The KGA results (3.2.3.2) only indicate that there are significant differences depending on the partner's income; It would be interesting to complete the section indicating whether these differences occur in all the subscales, in some items only, and if the association always occurs between the same levels (high score-high income, low score-high income, etc.).

6.- In addition, it is not indicated if there are differences with respect to other characteristics (level of education, partner’ level of education, age, etc.) or if they have not been analysed.

The discussion section is correct, the limitations of the work have been perfectly detected and highlighted, indicating future lines of research.

Author Response

Reviewer 2.

Reviewer’s comment: The introduction section is well justified and shows the relationship of the work with the relevant literature on IPV and its consequences. It correctly defines the objective of the study and the interest that a IVAW rapid detection questionnaire brings to healthcare professionals.

Response: Thanks for your positive feedback.

Reviewer’s comment: The materials and methods section includes the selection criteria and the sample sizes used. It is an online survey sent to social media platforms and women's associations, but,

1.- It does not indicate whether any response control strategy has been used, a characteristic that may influence the results obtained.

Response: Thanks for your comment. Although we included some technical control mechanisms to minimise bias regarding this matter (i.e. all the questions were compulsory to answer – please note that in the sociodemographic questions there was always the following response option “I would prefer not to disclose this information”- and the main researchers emails were provided for women to contact us in case they needed clarification), in hindsight we realise that we cannot guarantee that all the responders were women who at the time of completing the questionnaire had an intimate male partner.  We have included information about the technical control mechanisms implemented (see lines 96-101) and we have added the fact of not being able to ascertain that all respondents were women in an intimate relationship with a male as a limitation (see lines 293-295).

Reviewer’s comment: 2.- It would also be useful to indicate if the surveys have been answered with the questionnaire written in Spanish and the items have been subsequently translated; the psychometric results obtained will refer to the version used, and may be different in other languages.

Response: Thanks for your comment. Please see lines 102-103, where we have included such clarification.

Reviewer’s comment: In the pretest reliability study (2.5.2.2) it is indicated that to retain an item it must have an IT-CCI equal to or greater than 0.3; however,

3.- In Table 1, items 1 and 8 would not meet this criterion. It would be necessary to modify the criterion in section 2.5.2.2 (for example, IT-ICC equal to or greater than 0.2) or eliminate both items.

Response: Thanks for your comment. We have realised that the writing was confusing and did not reflect the criteria used. We have rewritten this section so it is clear that in order to remove an item, its IT-CCI must be lower than 0.3 AND the questionnaire’s Cronbach's α must increase significantly when that item is removed (see lines 134-137).

Reviewer’s comment: 4.- In the sociodemographic characteristics included in Table 2, it should be indicated whether the monthly income belongs to the person surveyed or to the household as a whole (the marital status of the people is not indicated either) since the economic dependence or independence can influence the results of perception of violence.

Response: Thanks for highlighting this. We have now clarified both in the text (see line 197) and the Table 2 that “monthly income” actually refers to “monthly women’s individual income”. Regarding the marital status of the participants, although we appreciate your point, when we design the study we considered that it was relevant to know how long women had been in a relationship and not necessarily whether they were single or married. Therefore, we cannot provide information as to whether they were married or not.

Reviewer’s comment: 5.- The KGA results (3.2.3.2) only indicate that there are significant differences depending on the partner's income; It would be interesting to complete the section indicating whether these differences occur in all the subscales, in some items only, and if the association always occurs between the same levels (high score-high income, low score-high income, etc.).

Response: Thanks for your comment. We have added a table (Table 4) summarising the results of comparing different groups regarding the partner’s income variable.

Reviewer’s comment: 6.- In addition, it is not indicated if there are differences with respect to other characteristics (level of education, partner’ level of education, age, etc.) or if they have not been analysed.

Response: Thanks for your comment. We have now clarified that no further differences amongst known-groups with respect to other variables were found (see lines 227-228).

Reviewer’s comment: The discussion section is correct, the limitations of the work have been perfectly detected and highlighted, indicating future lines of research.

Response: Thanks for your positive feedback

Reviewer 3 Report

Developing a measure of intimate partner violence for the context of Spain is important and novel. The only suggestion I have is to reflect on other validated IPV scales used previously and for many decades in other global contexts. What are their limitations and therefore the justification to create a new one? Are there adaptations needed for the culture/country context in Spain? 

Author Response

Developing a measure of intimate partner violence for the context of Spain is important and novel. The only suggestion I have is to reflect on other validated IPV scales used previously and for many decades in other global contexts. What are their limitations and therefore the justification to create a new one? Are there adaptations needed for the culture/country context in Spain?

Response: Thanks for your comment. Recent governmental efforts in Spain have led to a series of laws and regulations that intend to protect women against not only explicit forms of IPV but also more subtle forms of such violence. This has opened many public debates and it is contributing to raise the awareness about the importance of not accepting culturally imbedded behaviours that can be considered sexist microaggressions. In this context, we are at the point in which although explicit forms of IPV are widely condemned, IPV continues to be a national public health issue. We know that explicit forms of IPV are very often preceded by other more subtle forms of violence. We also know that at the basis of these forms of subtle violence are culturally accepted sexist behaviours. If we managed to assess when women are being subjected to invisible forms of sexist violence, we could be more likely to act early and avoid getting help for these women when it is too late. Most existing tools to assess IPV include items referring to very explicit forms of IPV, which can either lead to social desirability bias when completing them or to only detect very obvious forms of abuse. Many of these validated tools are validated in Spanish but they don’t help to detect invisible forms of IPV. The Q-IVAW intends to fill in the existing gap in the literature by helping healthcare professionals to detect when women are subjected to invisible forms of sexist violence from their partners.
